# The Singular Molecular Conformation of Humic Acids in Solution Influences Their Ability to Enhance Root Hydraulic Conductivity and Plant Growth

**DOI:** 10.3390/molecules26010003

**Published:** 2020-12-22

**Authors:** Maite Olaetxea, Veronica Mora, Roberto Baigorri, Angel M. Zamarreño, Jose M. García-Mina

**Affiliations:** 1Department of Environmental Biology, BIOMA Institut, Sciences School, University of Navarra, 31007 Pamplona, Spain; roberto.baigorri@tervalis.com (R.B.); angelmarizama@unav.es (A.M.Z.); 2Plant Physiology and Plant-Microorganism Interaction Laboratory, Universidad Nacional de Río Cuarto, Córdoba 5800, Argentina; moraveronica@yahoo.com.ar

**Keywords:** biomolecules, humic substances, polyacrylic acid, polyethylene glycol, plant growth improvement, molecular conformation

## Abstract

Some studies have reported that the capacity of humic substances to improve plant growth is dependent on their ability to increase root hydraulic conductivity. It was proposed that this effect is directly related to the structural conformation in solution of these substances. To study this hypothesis, the effects on root hydraulic conductivity and growth of cucumber plants of a sedimentary humic acid and two polymers—polyacrylic acid and polyethylene glycol—presenting a molecular conformation in water solution different from that of the humic acid have been studied. The results show that whereas the humic acid caused an increase in root hydraulic conductivity and plant growth, both the polyacrylic acid and the polyethylene glycol did not modify plant growth and caused a decrease in root hydraulic conductivity. These results can be explained by the different molecular conformation in water solution of the three molecular systems. The relationships between these biological effects and the molecular conformation of the three molecular systems in water solution are discussed.

## 1. Introduction

Humic substances (HS) may be considered as biomolecules formed in soil during the decomposition of plant and animal residues by chemical and biological processes [1] and partially resistant to microbial degradation [2]. It is well established that HS play important roles in the dynamics of metals and biomolecules (proteins, sugars) in natural ecosystems, mainly due to their complexing–binding abilities [3]. Likewise, HS affect both plant and microbe development in soils [3]. In this context, it becomes clear that adequate knowledge of those mechanisms that regulate these HS-mediated chemical and biological activities is of great importance to better understand the whole dynamics of natural or agronomic-related ecosystems.

Some studies have demonstrated that a sedimentary humic acid’s beneficial action on the growth of plants cultivated in hydroponics is related to its ability to affect root hydraulic conductivity and water root uptake [3,4,5]. Asli and Neumann [6] reported that the root application in maize of 1 g·L^−1^ (an unusual concentration in soil solution or agronomic practices) of both polyethylene glycol (PEG) and a sedimentary humic acid caused a reduction of shoot dry weight, root hydraulic conductivity, and leaf-transpiration rate. Other studies showed that when the humic acid concentration in solution was lower (0.1–0.3 g·L^−1^) and closer to that present in soil solution or involved in agronomic practices, its root application stimulated plant growth [3,7,8,9,10]. Further studies carried out using the range of humic acid concentration associated with increases in plant growth demonstrated that, in fact, the root application of a humic acid extracted from leonardite (HA) increased the root hydraulic conductivity very significantly [4]. These studies indicated that ABA signaling pathways regulated this process, and it involved the activation of some root aquaporins [4]. This action may cause a mild stress that activates plant responses to abiotic stress, thus reinforcing plants’ adaptation capacity to better growth under external stresses (priming action) [3,4]. Furthermore, the ability of HA to enhance plant growth was dependent on this increase in root hydraulic conductivity [4]. The hypothesis proposed in this study was that HA caused a transient fouling of root pores that disappeared with time due to conformational changes induced by HA’s interaction with the root surface [3,4].

To explore this hypothesis, we have compared the effect of HA with that of other molecular systems with similar hydrodynamic radii in solution but different molecular conformation, exploring the response to the root hydraulic conductivity and the growth of cucumber plants cultivated in hydroponics. Along with HA, we selected polyacrylic acid (PAA) and polyethylene glycol (PEG) molecular fractions with similar hydrodynamic radius (Rh) (around 1.5–4 nm) in solution to that of HA at the biological pH range (around 4 nm) and, therefore, presenting a potentially equivalent size in solution. It is interesting to note that these values of Rh are consistent with the average size (diameter) of root cell-wall pores in cucumber (around 6 nm) [11]. In principle, the three molecular systems have different molecular conformation in solution since PAA is a monodisperse acidic linear polymer, PEG is a monodisperse noncharged linear polymer, and HA is a polydisperse biomolecule formed through the self-association of different molecular units [3,12,13]. In this framework, it was assumed that the different molecular conformations in solution of the three molecular systems will be reflected in their action of root hydraulic conductivity and, thereby, on plant growth.

The effects on plant growth were also evaluated by measuring some of the principal markers of HA activity in plants, such as indolacetic acid (IAA) concentration in roots and cytokinin concentration (Cks) in leaves [14,15,16]. The potential water stress resulting from these molecules’ effects on root hydraulic conductivity was evaluated by measuring abscisic acid (ABA) and malondialdehyde (MDA) in leaves.

## 2. Results

### 2.1. Characterization of the Molecular Systems

The R_h_ (hydrodynamic radius) obtained by DLS (dynamic light scattering) for PAA and HA are presented in Table 1. The R_h_ for HA (4.1 nm) showed a value within the range of those for PEG (3.7 nm) and PAA (1.7 nm) (Table 1). Then, the MW for PAA was 30,000 Da, 20,000 Da for PEG, and finally, around 23,000 Da for the main size population of HA measured by DOSY ^1^H-NMR (Appendix A). Furthermore, we evaluated the total acidity and particle charge (molecular surface charge distribution, Z potential) since these variables are directly related to both the molecular conformation and potential chemical reactivity of the different macromolecules in solution. Regarding total acidity, PAA showed the highest acidity, 9.64 mmol·g^−1^ (Table 1); PEG showed the lowest, 0.01 mmol·g^−1^; and HA an intermediate value, 2.95 mmol·g^−1^, not as acid as PAA.

The evolution of Z potential in nutrient solution with pH was studied at three different pH values: pH 7, pH 6, and pH 4.5. These values of pH were selected in order to mimic a real pH environment in the rhizosphere influenced by the root PM- H^+^-ATPase activity. Results showed that PAA, the more acidic substance with particle charge well distributed around, presented less negative Z potential than HA (Figure 1). It was noteworthy that PEG, a noncharged polymer, presented more negative Z potential values than those of PAA (Figure 1). Regarding Z potential variation as a pH function, both PAA and HA showed slight Z potential variations. However, PEG showed an intense decrease in Z potential when pH increased from 6 to 7.

The three molecular systems were studied by fluorescence spectroscopy to determine their conformational structure (synchronous fluorescence spectra are presented in Appendix A). Following the argumentation given by Peuravuori [17], it is possible to relate a more complicated electronic structure with the “maxima” of the λ_ex_/λ_em_ displacement of the emission spectrum to longer wavelengths. The emission spectrum results for the three molecular systems yielded the shortest wavelengths for PEG (270/288 nm), followed by PAA (320/338 nm) (Figure 2). HA showed the longest wavelengths (320/338) (Figure 2). These results are related to the chemical nature and consequently to each substance’s electronic structure: PEG is a molecule without hydrolyzable functional groups, whereas PAA contains many carboxylic groups in its structure. Finally, HA presents phenolic and π-π electronic systems, showing the more complex electronic structure. The evolution of the emission spectrum maximum with the variation of pH values was different for each macromolecule. Thus, PAA underwent a high increase at pH 7, HA a slight increase at pH 7, and PEG did not show variations at any pH.

### 2.2. Biomarkers in HA-, PAA-, and PEG-Treated Cucumber Plants

Only HA-treated plants showed a statistically significant increase in both shoot dry weight (SDW) and root dry weight (RDW) (Table 2). Plants treated with PEG and PAA had SDW and RDW values similar to those of the control (Table 2). Besides, only HA-treated plants showed a statistically significant increase in IAA’s root concentration and the shoot concentration of active cytokinins (CKs) (Table 2). Regarding water relations of the plants, PAA- and PEG-treated plants showed a significant reduction in hydraulic conductivity (L_pr_) and stomatal conductance (Gs) when compared with control plants (Table 2) (the evolution over time of L_pr_ is included in Appendix A). Moreover, plants treated with both polymers showed an increase in ABA, a water stress indicator, and PEG-treated plants also showed an increase in MDA, an oxidative stress indicator. Conversely, HA-treated plants showed a significant increase in L_pr_, without variations in Gs, ABA, and MDA, compared with the control (Table 2).

### 2.3. Root Morphology Image after 72 h from the Onset of the Treatment

The snapshot of cucumber roots taken 72 h after the onset of the treatments was in agreement with the data presented for RDW (Figure 3). In this way, cucumber roots treated with HA showed higher root density, thicker roots, and more extensive secondary root development than for control, PEG-, and PAA-treated plants.

## 3. Discussion

The results concerning plant growth and physiology clearly show that both PEG and PAA do not behave as HA. Confirming previous studies [3,4,5,14,15,16], HA root application in cucumber caused a significant increase in both shoot and root dry weight compared to the control. In contrast, PAA and PEG did not cause any change in plant growth (Table 2). The effects on root growth were also reflected in the root morphology (Figure 3). In this way, HA-treated root plants showed higher root density and more considerable secondary root development compared to PEG- and PAA-treated plants. In agreement with previous studies [4], HA’s growth-promoting effect was accompanied by a concomitant increase in IAA root concentration and CK shoot concentration. However, neither PEG nor PAA treatments affected plant growth, root IAA, and shoot CK concentrations (Table 2).

These differences in plant growth and some plant hormones biosynthesis between the synthetic polymers and HA were also reflected in their effects on the plant’s L_pr_ and water relations. PEG and PAA caused a significant decrease in L_pr_ that was accompanied by an increase in ABA shoot concentration compared to control and HA-treated plants, which in turn caused a reduction in Gs and a decrease in leaf transpiration rates (Table 2). In the case of PEG, this water stress situation is associated with an increase in MDA, an oxidative stress marker (in the case of PAA, the increase was not significant). Conversely, in line with previous studies [4], the enhancement in plant growth associated with HA was linked to an increase in L_pr_ and did not affect ABA and MDA values in shoots (Table 2).

In principle, these three molecular systems’ different actions on plant growth and L_pr_ should be related to their structural conformation in solution. Molecular conformation is influenced by the ionic interactions between charged functional groups. Thus, repulsion forces associated with the interaction between ionized, negatively charged, oxygen-containing functional groups favor molecular expansion, whereas high ionic strength or low ionization favor molecular contraction involving H-bonds and attraction forces [12,13]. In this context R_h_, total acidity, and Z potential in solution are parameters of great interest.

As discussed above, R_h_ values of the three organic molecules in solution are within the same range (Table 1). Although these values corresponded to very different nominal MW values for each molecular system, their size distribution in solution (a component of the molecular conformation) is not very different. However, this fact is compatible with the presence of diverse whole molecular conformations in solution. Thus, PEG-20,000 yields unusually high MW when compared with globular proteins like human hemoglobin (MW, 68,000) or pepsin (MW, 40,700) [18]. More recently, PEG conformation was estimated as random chains presenting significant anisotropy and giving PEG axes like prolate or oblate spheroids [19] (Figure 4). As for PAA, simulations of PAA conformation changes from low to high ionic strength showed a variation between a torus and a helical shape, respectively [20] (Figure 4). Finally, humic acids have shown a more complex and flexible behavior in solution depending on pH, HA concentration, or ionic strength. The application of an extended polyelectrolyte model to PAA and HA’s acid–base behavior showed that HA conformation was much more sensitive to pH and ionic strength changes than PAA [21]. Thus, HA could form extended random coils at neutral pH but more collapsed structures at a low pH, and probably spherical shapes by micelle formation at high concentration [22] (Figure 4). High ionic strength values could also have this effect on HA conformations [22].

The above-mentioned molecular shapes corresponding to each molecular system are described in Figure 4.

Likewise, pH changes can also be associated with molecular aggregation and disaggregation phenomena in HA solutions [12,13].

As mentioned above, these differences in the potential conformational behavior ascribed to each molecular system are associated, at least to some extent, with the molecular electric charge distribution, and charge–charge attractive or repulsive interactions, as a function of pH and counter-ions (ionic strength). It is noteworthy that although the total acidity of PAA is higher than that of HA (Table 1), Z potential values (indicative of the charge distribution on the molecular surface) are less negative than that of HA (Table 1). This result is consistent with the possible micelle formation in the HA system, with negative ionized functional groups oriented and grouped in the outer side of the molecular surface, while more hydrophobic molecular regions are grouped inside. However, in PAA, the negative charge distribution seems to be more homogeneously distributed in the whole molecule. Probably, this difference between HA and PAA significantly affects the chemical and biological features of both molecular systems, since charge density on the molecular surface plays a very important role in ion attraction and further binding in the molecular system, and molecular system–root cell wall interaction as well. In this sense, it is quite relevant that a noncharged polymer as PEG presented significant negative Z potential values, which were even higher than those of PAA. This fact, which is probably linked to O-induced dipoles distributed throughout the PEG structure, also shows that PEG-root pore interaction probably involves not only steric factors but also electrostatic ones. On the other hand, the results also show that Z potential values did not change significantly in this pH range for the three molecular systems (Figure 1). This fact indicates that within this range of pH, the three macromolecules’ molecular conformation could be relatively stable.

Fluorescence study also supplies interesting information about the structural features of the three molecular systems. The shorter emission wavelength for PEG indicates a simpler electronic configuration on fluorophore groups, while emission wavelength for PAA indicates a more complicated electronic configuration. HA, with the largest wavelength, is related to the most complicated electronic features (Figure 2 and Appendix A). Also, the pH evolution of fluorescence maximum (Figure 2) and synchronous spectra (Appendix A) indicates an electronic complexity of HA moieties much higher than that of PAA and PEG. These results, along with the presence of more types of acidic groups in HA (carboxylic and phenolic), indicate that the molecular complexity of HA moieties is much higher than that of PAA and PEG. This fact is likely due to structural regions with high aromatic character in the HA structure or HA molecular conformation

These results clearly show that the three molecular systems behave differently in solution, with HA presenting more flexible molecular conformations that might include molecular rearrangements due to electrostatic interactions. These different molecular conformations are probably influencing their interaction with pore cells. Both PAA and PEG, which presented more static molecular conformations, seem to remain inside pore cells, causing a stable fouling. However, HA, which presents more molecular flexibility and some degree of molecular aggregation, might cause a transient fouling of pore cells due to molecular rearrangements associated with the HA interaction with root cell membrane-related compounds such as root exudates and protons.

In summary, the results obtained in this study support the hypothesis that the molecular conformation in solution at the root surface of HA plays a crucial role in modifying root hydraulic conductivity and plant growth. However, further studies must be carried out in order to elucidate what type of conformational changes are involved in this process and if these include molecular rearrangements (for example, molecular disaggregation).

## 4. Materials and Methods

### 4.1. Characterization of HA, PAA and PEG

The sample of HA used in the experiments was obtained from leonardite. A specific amount of HA (100 g) was extracted, and purified using the IHSS methodology (http://www.humicsubstances.org/soilhafa.html) as described in previous work [14]. The concentration of the main phytoregulators in HA composition was assessed employing high-performance liquid chromatography/mass spectrometry (HPLC/MS) as described previously [14]. Finally, HA was characterized using ^13^C nuclear magnetic resonance (^13^C-NMR), high-performance size exclusion chromatography (HPSEC), and elemental analysis as described in background works [14]. The HA molecular weight distribution and R_h_ were also studied using DOSY ^1^H-NMR and DLS, respectively (Appendix A).

### 4.2. Acidity or Functional (Potentiometry) Analysis

Measures were carried out on solutions prepared by dissolving an adequate amount of material in 0.1 M NaOH. Once the material had been dissolved, an H^+^-cationic exchange resin (Amberlite IRA-118H^+^) was added to the stock solution to attain a final pH of 3.5. The resin was then separated by centrifugation (15 min at 5000× *g* of centrifugal force). In order to carry out the titration studies, an aliquot of the stock solution corresponding to 50 mg of the polymer system was added to a water solution containing 0.5 mL of 0.1 M HClO_4_ and the required volumes of 1 M KNO_3_ for fixing ionic strength (I) values (0.01 M). The final volume was 35 mL. The solution was titrated with 0.05 mL increments of 0.1 M NaOH by using a Metrohm Titrando 809 under N_2_ atmosphere. The pH was registered using a combined pH glass electrode of the same company. To ensure that equilibrium between measurements was reached, no base was added until the pH measurement remained stable with a variation of pH no greater than 0.01 pH unit over 5 min. The experimental data were treated following the analysis of functional groups described in previous work [23].

### 4.3. Z Potential and Dynamic Light Scattering Measurements

Analyses were carried out at work concentration in nutrient solution conditions on a Zeta Plus (Zeta Potential Analyzer) from Brookhaven Instruments Corporation using an AQ-765 cell (750 Blue Point Road, Holtsville, NY 11742, USA). Measures were the average of thirty repetitions.

Dynamic light scattering (DLS) measurements were made at a scattering angle of 90° and at 25 °C using a DynaPro-MS/X photon correlation spectrometer (Bluefactory, Bât, A Passage du Cardinal 1 CH-1700 Fribourg, Switzerland), equipped with a 248-channel multi-tau correlator and a Peltier effect thermostabilization unit. The wavelength of the laser was 825.5 nm. The size distribution was obtained from the intensity autocorrelation function by regularization analysis, implemented in the DynamicsTM software package, and the hydrodynamic radii (R_h_) calculated from measured diffusion coefficients by means of the Stokes-Einstein equation (Equation (1)):R_h_ = kT/6πη_0_D_0_(1)
where k is the Boltzmann constant, T the absolute temperature, η_0_ the solution viscosity, and D_0_ is the diffusion coefficient at zero concentration. All samples were filtered through 0.45 mm nylon filters before analysis.

### 4.4. Fluorescence Spectroscopy

Measures were performed at work concentration in nutrient solution on a PerkinElmer (Waltham, MA, USA) LS50B fluorescence spectrophotometer, under the same conditions described in [17]. Briefly, samples were dissolved (50 mg·L^−^^1^) into sodium acetate buffer (pH 7.0). Emission spectra were collected in the 250–600 nm wavelength range at the maximum of excitation for each sample. All spectra were recorded with a 5 nm slit width on both monochromators. The scan speed of spectra was 120 nm·min^−^^1^ and resolution 0.5 nm.

### 4.5. Plant Material and Culture Conditions

Seeds of cucumber (*Cucumis sativus* L. cv Ashley) were germinated in water with 1 mM of CaSO_4_, in darkness, on perlite and moistened filter paper in a seed germination chamber. One week after germination, plants were transferred to 8 L recipients in hydroponic solution. The nutrient solution used was: 0.63 mM K_2_SO_4_; 0.5 mM KH_2_PO_4_; 0.5 mM CaSO_4_; 0.30 mM MgSO_4_; 0.25 mM KNO_3_; 0.05 mM KCl and 0.87 mM Mg(NO_3_)_2_; 40 µM H_3_BO_3_; 4 µM MnSO_4_; 2 µM CuSO_4_; 4 µM ZnSO_4_ and 1.4 µM Na_2_MoO_4_. The nutrient solution contained 40 µM of iron as EDDHA chelate (80% ortho-ortho isomer). No precipitation of Fe inorganic species was observed throughout the experiment. The pH of the nutrient solutions was held at 6.0 and did not change significantly during the experiment. All experiments were performed in a growth chamber at 28/21 °C, 70–75% relative humidity, and with 15/9 h day/night photoperiod (irradiance: 250 µmol·m^−2^·s^−1^). After 10 days of plant growth, the following treatments were carried out: a control treatment that only received the nutrient solution, HA treatment, PEG treatment and PAA. Harvests were conducted at the same time of the day to exclude diurnal variations, which meant 6 h after the start of the light period. Plants were harvested at 4, 24, 48, and 72 h after the application of HA and PEG and PAA treatments. One part of the plant material was weighed and dried (60 °C) for shoot dry weight (SDW) and root dry weight (RDW) determination, and another part was frozen in liquid nitrogen and stored at −80 °C for further analysis. All determinations were carried out employing five replications.

### 4.6. Treatments with HA, PAA, and PEG

For HA treatments, a concentration of 100 mg·L^−^^1^ (expressed as organic carbon) of HA was supplied to nutrient solution as HA treatment. Polyacrylic acid (PAA 30,000 Da) and polyethylene glycol (PEG 20,000 Da) reagents were purchased from Sigma-Aldrich. Both of them were used at the same concentration as that of HA.

### 4.7. Measurement of Stomatal Conductance (Gs)

Stomatal conductance at 72 h was measured in cucumber leaves. A portable photosynthesis measuring system CIRAS-2 (PPSystems, Hitchin, UK) was fitted with broad leaf cuvette, at irradiance of 303 µmol·m^−^^2^·s^−^^1^, 405 cm^3^·m^−^^3^ CO_2_ in air, and leaf temperature of 26 °C.

### 4.8. Analysis of the Concentration of Phytoregulators in Plant Tissues

The concentration of the principal plant regulators was analyzed using HPLC/MS/MS as described below.

The following hormones were studied: trans-zeatin (t-Z), trans-zeatin riboside (t-ZR), and isopentenyladenine (iP) as citokinins (CK), indol acetic acid (IAA), and abscisic acid (ABA). The extraction and purification of the different plant regulators were carried out using the previously described methods [15].

Liquid chromatography–mass spectrometry quantification of CKs: the CKs were quantified by HPLC linked to a 3200 Q TRAP LC/MS/MS system (Applied Biosystems/MDS Sciex, Ontario, ON, Canada), equipped with an electrospray interface, using a reverse-phase column (Tracer Excel 120 ODSA 3 µm, 100 × 4.6 mm, Teknokroma, Barcelona, Spain). A linear gradient of methanol and 0.05% formic acid in water was used: 35–95% methanol for 11 min, 95% methanol for 3 min, and 95%–35% methanol for 1 min, followed by a stabilization time of 5 min. The flow rate was 0.25 mL/min, the injection volume was 50 µL, and the column and sample temperatures were 30 and 20 °C, respectively.

Detection and quantification were performed by multiple reaction monitoring (MRM) in the positive-ion mode, employing a multilevel calibration graph with deuterated CKs as internal standards. The source parameters were: curtain gas: 25.0 psi, GS1: 50.0 psi, GS3: 60.0 psi, ion spray voltage: 5000 V, CAD gas: medium, and temperature: 600 °C.

Liquid chromatography–mass spectrometry quantification of ABA: the hormone was quantified by HPLC linked to a 3200 Q TRAP LC/MS/MS system (Applied Biosystems/MDS Sciex, Ontario, ON, Canada), equipped with an electrospray interface, using a reverse-phase column (Synergi 4 µm Hydro-RP 80A, 150 × 2 mm, Phenomenex, Torrance, CA, USA). A linear gradient of methanol and 0.5% acetic acid in water was used: 35% methanol for 1 min, 35–95%methanol for 9 min, 95% methanol for 4 min, and 95%–35% methanol for 1 min, followed by a stabilization time of 5 min. The flow rate was 0.20 mL/min, the injection volume was 50 µL, and the column and sample temperatures were 30 and 20 °C, respectively.

Detection and quantification were performed by MRM in the negative-ion mode, employing a multilevel calibration graph with deuterated hormones as internal standards. The source parameters were: curtain gas: 25.0 psi, GS1: 50.0 psi, GS3: 60.0 psi, ion spray voltage: −4000 V, CAD gas: medium, and temperature: 600 °C.

### 4.9. Measurement of Root Exudation in the Absence of Hydrostatic Pressure Gradients (Osmotic Exudation) (L_pr_)

Root exudates collection: the collection of xylem sap is based on “root pressure” and water follows a “cell-to cell” pathway [24,25]. Plants were excised in their growing hydroponic medium. Stems were first cut from just below the first leaf. Then, the top part of the stem was introduced in a silicone tube and sealed with a self-sealing film to avoid any loss of sap. Root exudate was finally collected with a glass Pasteur pipette. Collections were done continuously during the first 90 min of exudation at 4, 24, 48, 72 h of treatment and kept in a previously weighed 1.5 mL tube.

Measurement of the osmotic pressure of exuded sap: the osmolality of root exudates was measured using a freezing point depression osmometer (Osmomat 010 Gonotec, Germany). Osmolality (mOsmol/kg) was converted to osmotic pressure (MPa) according to a described procedure [26]:MPa = mOsmol × 0.831 × 10 − 5 × T(°K)(2)

In the absence of hydrostatic pressure gradients, water uptake by the root is governed by the differences in osmotic pressure between the medium and the sap [27]. The equation to calculate hydraulic conductivity (L_pr_) is described as:J_v_ = L_pr_ × σ_sr_ × Δπ (3)

The coefficient σ_sr_ denotes the reflection coefficient of solutes in the roots, which was reported to be 0.853 by previous studies [28]. Thus, we could measure root hydraulic conductivity (L_pr_) by measuring water flow and pressure differences (Δπ).

Results were calculated by means of two independent experiments and three replications in each experiment.

### 4.10. Determination of Malondialdehyde (MDA) Concentration in Leaves

The MDA concentration was measured following a described methodology [29]. A 0.4 g amount of frozen plant material was homogenized in 5 mL of 80% cold ethanol using a tissue. Homogenates were centrifuged at 4 °C to pellet debris and different aliquots of the supernatant were mixed either with 20% trichloroacetic acid (TCA) or a mixture of 20% TCA and 0.5% thiobarbituric acid (TBA). Both mixtures were allowed to react in a water bath at 90 °C for 1 h. After this time, samples were cooled down in an ice bath and centrifuged. Absorbance at 440, 534, and 600 nm was read in the supernatant against a blank. The MDA concentration in the extracts was calculated as follows:[(Abs 532 + TBA) − (Abs 600 + TBA) − (Abs 532-TBA − Abs 600 − TBA)] A[(Abs 440 + TBA − Abs 600 + TBA) × 0.0571] BMDA equivalents (nmol·mL^−1^) (A − B/157 000) × 106

The MDA concentration was expressed as nmol MDA per gram of fresh weight (FW).

### 4.11. Root Morphology Images

Root Morphology Images Were Taken with a Bridge Camera (Canon PowerShot SX540)(Bovenkerkerweg 59, 1185 XB, Amsterdam, The Netherlands). Cucumber roots were submerged in distilled water in order to take the photographs.

### 4.12. Statistical Analysis

Significant differences (*p* < 0.05; 0.01) among treatments were calculated by using one-way analysis of variance (ANOVA) and the LSD Fisher post hoc test. All statistical tests were performed using the statistical package Statistica 6.0 (StatSoft, Tulsa, OK 74104, USA).

## Figures and Tables

**Figure 1 molecules-26-00003-f001:**
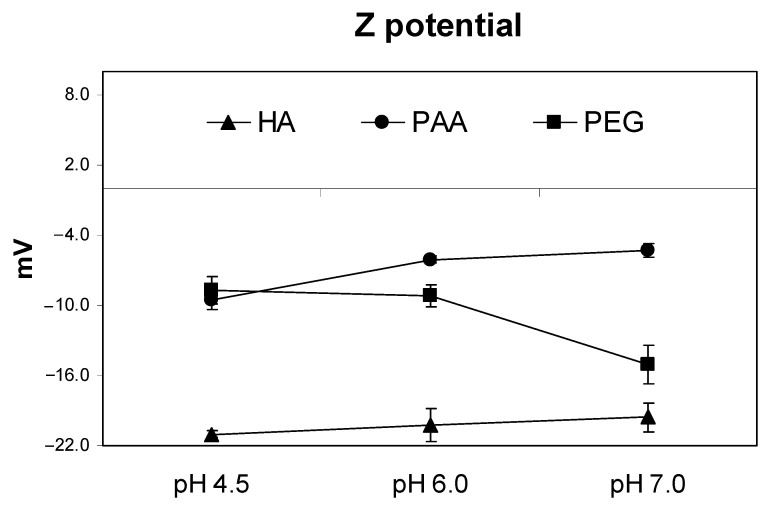
Evolution at different pH of Z potential for PEG, PAA, and HA at work concentration solutions. The measurements correspond to the mean value (*n* = 3) ± standard deviation.

**Figure 2 molecules-26-00003-f002:**
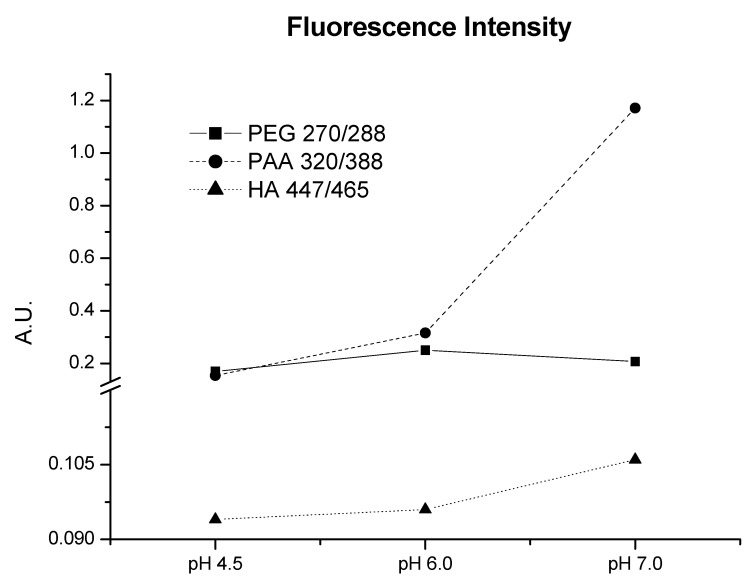
Variations at different pH of the emission/excitation peak fluorescence maximum for PEG, PAA, and HA at work concentration solutions. Intensities expressed in arbitrary units (A.U.).

**Figure 3 molecules-26-00003-f003:**
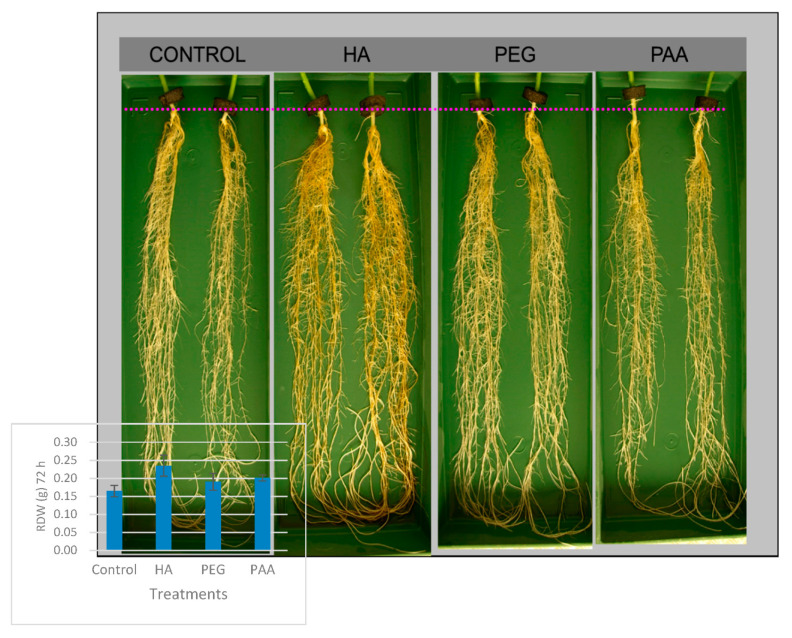
Picture of the growth of cucumber roots after 72 h of treatment.

**Figure 4 molecules-26-00003-f004:**
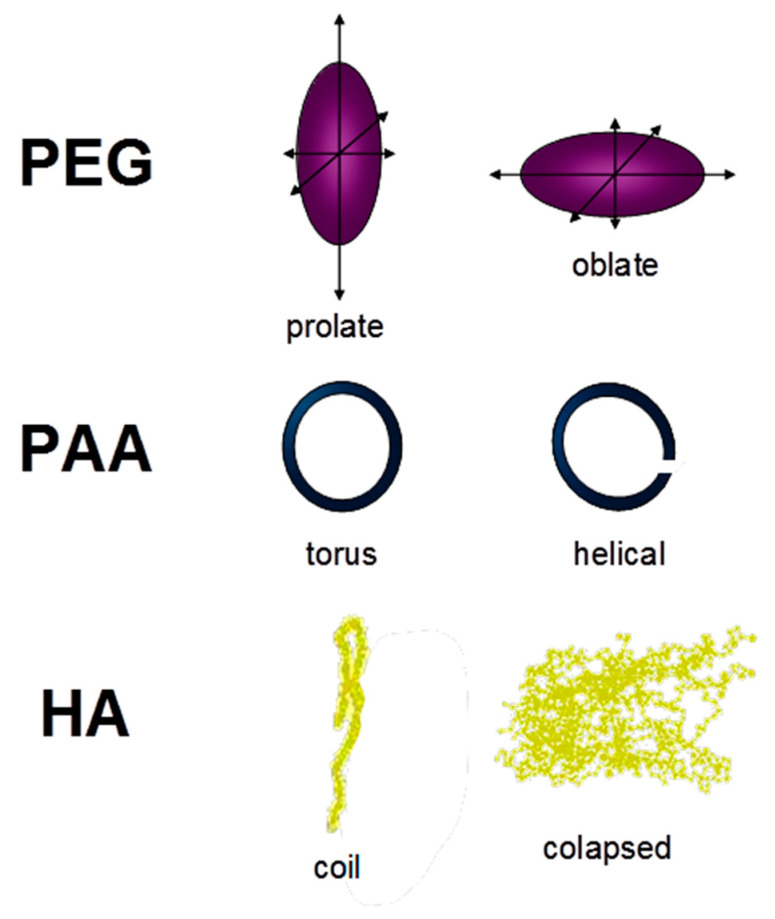
Different shapes proposed for PEG, PAA, and HA. PEG conformation was estimated as random chains presenting significant anisotropy and giving PEG axes like prolate or oblate spheroids. PAA conformation changes from low to high ionic strength showed a variation between a torus and a helical shape. HA could form extended random coils at neutral pH but more collapsed structures at low pH.

**Table 1 molecules-26-00003-t001:** Hydrodynamic radius, acidity, and charge (Z potential) of the different macromolecules in solution at work concentration and pH. The values correspond to the mean value (*n* = 3) ± standard deviation.

	R_h_ (nm)	Acidity (mmol·g^−1^)	Z Potential
HA	4.1 ± 0.04	2.95 ± 0.05	−20.2 ± 1.43
PAA	1.7 ± 0.13	9.64 ± 0.08	−6.07 ± 0.34
PEG	3.7 ± 0.05	0.01 ± 0.00	−9.16 ± 0.90

**Table 2 molecules-26-00003-t002:** Biomarkers content for different treatments applied.

	SDW	RDW	L_pr_	G_S_	IAA	Cks	ABA	MDA
CONTROL	0.65 ± 0.04	0.16 ± 0.02	24.8 ± 2.2 *	451 ± 29.9 *	55.5 ± 1.00	0.75 ± 0.14	39.2 ± 4.61	12.7 ± 1.74
HA	0.86 ± 0.09 *	0.23 ± 0.03 *	40.7 ± 10 **	414 ± 54.0 *	62.0 ± 2.04 *	1.04 ± 0.20 *	39.8 ± 5.01	12.1 ± 1.03
PEG	0.70 ± 0.08	0.19 ± 0.02	14.2 ± 2.0	364 ± 21.2	50.1 ± 1.18	0.61 ± 0.08	54.9 ± 4.06 *	15.7 ± 2.14 *
PAA	0.76 ± 0.06	0.20 ± 0.02	11.5 ± 4.9	270 ± 46.2	53.7 ± 1.89	0.65 ± 0.14	55.6 ± 8.87*	13.3 ± 1.59

(*) Significant differences (*p* < 0.05). (**) Significant differences (*p* < 0.01). Measures at maximum activity time for each parameter (*n* = 5) ± standard deviation: SDW and RDW after 72 h; IAA after 48 h in roots; L_pr_ after 72 h in root, Gs after 72 h in leaves; Cks after 48 h in leaves; ABA and MDA after 72 h in leaves. SDW and RDW in g; L_pr_ in g H_2_O g^−1^ DW MPa^−1^·h^−1^; IAA in pmol·g^−1^ FW; Gs in mmol m^−2^·s^−1^; CK in pmol·g^−1^ FW; ABA in pmol·g^−1^ FW; MDA in nmol·g^−1^ FW. These experiments were replicated three times.

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
