# Peer review of "The Singular Molecular Conformation of Humic Acids in Solution Influences Their Ability to Enhance Root Hydraulic Conductivity and Plant Growth"

_molecules, 2020, doi:10.3390/molecules26010003_

Round 1

Reviewer 1 Report

Olaetxea et al. compared in this concise manuscript the growth-enhancing effect between humic acid (HA), polyacrylic acid (PAA) and polyethylene glycol (PEG) solutions. The authors found that, despite having similar hydrodynamic radii (Rh), the application of the three molecules can lead to different growth effects in cucumber, namely that HA enhances growth whereas PAA and PEG reduce growth. The authors later subjected the cause to different molecular conformations.

Whereas the topic and message of the manuscript is interesting, the data representation can use some improvement. Particularly, data in both Table 1, 2 and Figure 2 do not contain any form of measurement uncertainty, and information on experimental repeats should be incorporated into legends. Similarly, Figure 1 also misses explanation of the error bars. The authors can also consider expanding upon the data acquired for more insight: do the osmolality and sap flux change over time after excision (the authors have several time points)? Did the root structure change in association with dry weight and Lpr change? Etc. A more comprehensive analysis of the data would have been appreciated.

To sum up, the manuscript misses key statistics and can use some improvement in data representation and interpretation. With that said, unfortunately I cannot recommend acceptance of the manuscript in this state.

Author Response

Reviewer 1:

Data in table 1 and table 2: standard deviation errors included.

Figure 2: Fluorescence spectra data are usually represented without the technical error.

Evolution of the hydraulic conductivity over time: we included a figure in Supplementary Information data (Figure S4) showing the evolution over time of Lpr under the plant treatments.

We included new data to analyze the influence of the treatments in the general root structure (figure 3).

Explanations of the type of error bars, the number of repetition for each measurement and the number of experiments repeated are included in the legends of each table and figure.

Reviewer 2 Report

I feel there is lack of novelty, especially there is published part in the beginning of the results and discussion part??

The size and strength of the results is limited 

I feel it needs additional data to be added.

The authors should add the significant differences on figures 1 and 2 as well as in other results.

Author Response

Reviewer 2:

Although we have some other publications in which we showed the mechanism of action of HA based on the response of some plant physiological biomarkers (Olaetxea et al. 2015; Olaetxea et al. 2019), the experiments carried out for this publication are original.  In fact, the perspective of this work was to study more in depth the relationship between the structural conformation of the biomolecules and the physiological response in plants (this is the reason why in our experiments we compare the behaviour of HA, PEG and PAA).

We included the error bars in the data represented in the different tables and figures.

Reviewer 3 Report

In this paper, the authors evaluated effects of humic acid, polyacrylic acid and polyethylene glycol in cucumber growth and root hydraulic conductivity. They found that humic acid increased the root hydraulic conductivity and whole plant growth, whereas other two molecules decreased the root hydraulic conductivity and did not affect the plant growth. Humic acid is a bioactive biostimulant, recently focused by many plant biologists. The results might be informative for the readers. However, the reviewer considers the paper should be improved before the publication in molecules.

1) Abbreviations should be defined. L. 84, DLS, L.136, Lpr and Gs. In Materials and Methods section, they are defined, but the authors should define them when they are used in the first time.

2) Information of the figure legends and Tables are very poor. Are the error bars S.D. or S.E.? How many experiments did the authors repeat? How many samples did the authors use in each experiment? In table 2, what is C? Control?

3) Figure 3, The figure is just repetition of the other data. Please rather provide photographs that the cucumber growth is presented.

4) Table 1, L. 97, What is #[17]? No significant difference among HA, PAA, and PEG?

5)The authors evaluated effects of pH 4.5, 6.0, and 7.0 in the experiments. Why did they choose the pH? Please explain it in the text.

6) Figure 4, the reviewer could not follow what the authors want to show because of no explanation in the legened.

7) Author affiliations, No country name. Is it OK?

Author Response

Reviewer 3:

1)  Abbreviations included in the results section when they are used for the first time.

2) Information of the legends extended. The information about the number of experiments included. Error bars in data tables and figures corrected.

3) We removed figure 3 from the text because it was included as the article icon. However, we included another new figure showing the influence of the treatments in root architecture and growth.

4) [# 17] corresponded to a literature cite. In this new version, we include the real data of the Rh for PEG. We omitted the literature cite.

5) The reason why we selected these pH values is included in the text (line 102).

6) Explanations of figure 4 included.

7) Country name included in author affiliations.

Round 2

Reviewer 2 Report

Accepted 

Reviewer 3 Report

In this time, I do not have any suggestions.